# Linear Relational Decoding of Morphological Relations in Language Models

## Abstract

The recent success of transformer language models owes much to their conversational fluency and productivity in linguistic and morphological aspects. An affine Taylor approximation has been found to be a good approximation for transformer computations over certain factual and encyclopedic relations. We show that the truly linear approximation $W\mathbf{s}$, where $\mathbf{s}$ is a middle layer representation of the base form and $W$ is a local model derivative, is necessary and sufficient to approximate *morphological derivations*. This approach achieves above 80% faithfulness across most morphological tasks in the Bigger Analogy Test Set, and is successful over different language models and languages. We propose that morphological relationships in transformer models are likely to be linearly encoded, with implications for how entities are represented in latent space.

## 1 Introduction

Large language models display impressive capabilities for factual recall, which commonly involve relations between entities (Brown et al. 2020). Recent work has shown that affine transformations on subject representations can faithfully approximate model outputs for certain subject-object relations (Hernandez et al. 2023). Identifying the contexts in which approximations perform well is an important area of study, with applications in interpretability and model editing.

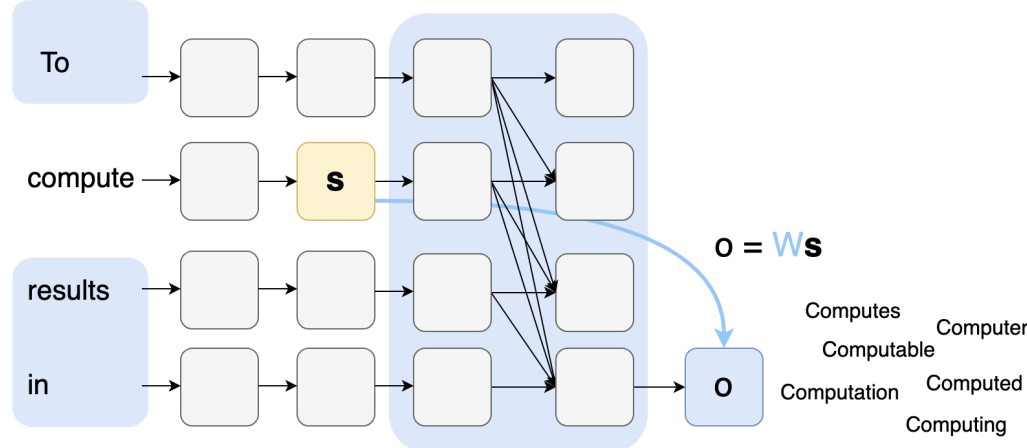

Figure 1: Adapting morphological analogies from the Bigger Analogy Test Set to relational contexts reveals that many are uniquely linearly approximable from base forms, such as **[verb+tion_irreg]**, **[verb+able_reg]**, **[noun - plural_reg]**, **[verb_inf - Ved]**, and **[verb+er_irreg].**

Work to date around relational representation in LMs have primarily focused on relations in the context of factual subjects and objects Meng et al. (2022b), Hernandez et al. (2023), Chanin et al. (2023). However, relations in natural language encompass a much broader range of subject and

object relations. Much of the mainstream success of LLMs has been due to the conversational nature of chat-oriented language models. The impressive conversational ability of LLMs depends on their linguistic competency, including lexical and morphological productivity, and uncovering how models are able to achieve this is an important aspect of model interpretability.

We make the assumption that *morphology* constitutes a portion of a model's relational knowledge. That is, we create an approximator which maps base representations ('compute') $\mathbf{s}$ to its corresponding output representations ('computable'), as $F(\mathbf{s}) = W\mathbf{s}$, similar to the $LRE(\mathbf{s}) = \beta W_r \mathbf{s} + b_r$ seen in Hernandez et al. (2023) but omitting the bias term $b_r$ and hyperparameter $\beta$. We show that the Jacobian transformation effectively approximates object decoding from an enriched subject state in morphological relations, often surpassing the affine method.

With this simple linear approximation, we find that approximable relations include pluralization, nominalization, changes in tense, and resultative forms. These derivations range over different parts of speech, including noun to adjective **[noun+less]**, adjective to noun **[adj+ness]**, verb to noun **[verb+er]**, and verb to adjective **[verb+able]**, and involve diverse subjects and objects. Importantly, we find that relative to linear approximation, *encyclopedic* and *semantic* relations benefit from the affine LRE, but not *morphological* relations.

Our study reproduces and extends existing research. Specifically, we apply the affine Linear Relational Embedding (LRE) method to novel relational categories encompassing a diverse range of domains, including derivational and inflectional morphology, encyclopedic knowledge, and lexical semantics. We address the data scarcity of the original paper in many categories, and confirm the efficacy of the affine LRE. We show that relational approximation can be applied to an adapted *analogical dataset* and demonstrate relational approximation for a broad range of linguistic phenomena. This opens avenues for further research in relational representations which take advantage of existing analogical and linguistic datasets, such as MarkG, SCAN, WordNet, and FrameNet (Zhang et al. 2022, Miller 1995, Czinczoll et al. 2022, Baker et al. 1998).

At the same time, it makes a key contribution to the body of research around relational representation in model latents. We show that for different relations, additive and multiplicative mechanisms play complementary roles in affine approximation. We find that the original linear relational embedding developed by Paccanaro and Hinton (2001), a multiplicative operator, is effective within specific relations. In particular, linear approximation within contexts relating morphological forms reaches near-equivalent level of faithfulness to the approximation found by the affine LRE. We perform tests in eight different languages, and find that this equivalence holds across typological categories.

## 2 RELATED WORK

Much work in machine learning has focused on learning concept representations with hierarchical structure. Relations between representations in concept spaces have been modeled successfully by both linear multiplicative and additive operations.

### 2.1 LINEAR EMBEDDING SPACES

**Multiplicative.** Paccanaro and Hinton (2001) introduced the concept of the linear relational embedding for learning relational knowledge from triples $(a, R, b)$. Along with prior work (Hinton 1986), they were able to solve a family tree problem where data is given in relational triples (Colin, *child*, Victoria), where vector components captured implicit semantics such as generation. Concepts such as $a$ and $b$ are represented as $n$-length vectors, while relations such as $R$ are represented as $n \times n$ matrices, akin to Coecke's models of compositional semantics (2010).

**Additive.** Mikolov et al. (2013) used linear operations in word vector space derived from context-predictive neural nets, demonstrating a correspondence between directional binary relations (male-female, country-capital, verb tense) and the addition of certain embedding vectors. Subsequent work with GloVe focused on leveraging statistical information to develop semantic substructures e.g. in Pennington et al. (2014). Detailed empirical studies found inflection relations (*comparative*, strong:stronger) are better captured than derivation relations (*lacking*, life:lifeless), and that encyclopedic relations (*capital-of*, Greece:Athens) are better captured than lexicographic relations (*member-of*, player:team) (Gladkova et al. 2016; Vylomova et al. 2016).

Park et al. 2023 formalize the compositional representation of concepts in embedding spaces. Extending prior work (Wang et al. 2023), they define a set of counterfactual outputs $Y$ for a directional binary concept $W$. They identify concept intervention as adding an embedding representation $\bar{\lambda}_W$ to change the probability of an output reflecting a concept $W$. For any concept $Z$ linearly separable from $W$, an output word $Y(W, \ldots, Z)$, and concept embedding $\lambda$, an intervention is effective if it changes the probability of $W$ but not $Z$.

## 3 BACKGROUND

### 3.1 TRANSFORMER COMPUTATION

In auto-regressive transformer language models, input text is converted to a sequence of tokens $t_1 \ldots t_n$, which are subsequently embedded as $x_1 \ldots x_n \in \mathbb{R}^d$ by an embedding matrix. They are then passed through $L$ transformer layers, each composed of a self-attention layer and an multi-layer perceptron (MLP) layer. In GPT-J, the representation $x_i^l$ of the $i^{\text{th}}$ token at layer $l$ is obtained as:

$$x_i^l = x_i^{l-1} + a_i^l + m_i^l$$

where $a_i^l$ is multi-headed Key-Value Query attention over $x^{l-1}$(Vaswani et al. 2017) and $m_i^l$ is the $i^{\text{th}}$ output of the $l^{\text{th}}$ MLP sublayer. Note that the output of the $l$-th MLP sublayer for the $i$-th representation depends on $x_i^{l-1}$, rather than $a_i^l + x_i^{l-1}$ (Wang and Komatsuzaki 2021). A decoder head $D$ consists of a linear layer and softmax to a token vocabulary. The final token prediction $t_{n+1}$ is then determined by $D$, applied to the contextualized final state corresponding to the token $t_n$:

$$t_{n+1} = \underset{t}{\operatorname{argmax}} \, D(x_n^L)_t$$

Throughout this paper, we will focus solely on subject-object relations, as expressed through a single relation (e.g. *Miles Davis plays the trumpet*). Following the insights of Meng 2022b and Geva 2023 that the last subject token state in middle layers are strongly casual on predictions, we are interested in utilizing the gradient between the last token position of the subject $s$ at an intermediate layer, and the last token position overall, the prediction $o$. We will refer to the middle layer final subject token state as **s**, and the final object token state as **o**.

### 3.2 INTERNAL RELATIONAL REPRESENTATION

In recent years, there has been increasing interest in factual relational representation. Meng et al. (2022b) found that factual statement predictions exhibit strongly causal states in middle layers at last subject token, supporting the idea that an enriched subject representation exists prior to output. Geva et al. (2023) demonstrated that attribute extraction is often performed by specific attention heads in later layers, and takes the form of a query on the enriched representation.

We directly build off of work by Hernandez et al. 2023, who present an approximator known by the corresponding internal hypothesis of the Linear Relational Encoding. Within this paper, we will denote this as the *affine LRE*. With $s$ denoting a middle hidden subject state and $o$ denoting the final object state, they treat object-retrieval for a relational context $r$ as linearly approximable: $o = F_r(s)$. They model $o$ with an affine first-order Taylor approximation

$$o = F_r(s) \approx Ws + b$$

using the transformer Jacobian $\frac{\partial F}{\partial s}$ between states to approximate $W$, and utilizing the subject representation $s$ from an intermediate layer. By doing so, they achieve over 60% faithfulness for LM predictions across certain factual, commonsense, linguistic, and bias relations.

In this paper, we identify a linear relational embedding mechanism for morphological relations. Through coarse-grained methods such as linear probing, transformer models have been found to encode linguistic features in internal representations, such as syntactic dependencies and thematic categories (Kann et al. 2018; Tenney et al. 2019; Wilson et al. 2023). Lin et al. (2019) identifies aspects of syntactic structure that are relevant for subject-verb agreement and reflexive dependencies in BERT. However, there remains a lack of explicitly identified encoding mechanisms for many linguistic competencies observed in language models, including morphology, grammatical agreement, ambiguity resolution, and discourse coherence.

## 4 APPROACH

### 4.1 PROBLEM STATEMENT

Approximating transformer computations through affine approximation has achieved empirical success, yet the internal mechanism by which they operate remain opaque. Beyond whether the outputs of an language model are approximable from latent representations, we would like to gain insight as to how the underlying relationships between concepts are represented. Within contexts which express relations, we would like to understand if models implement simple, linear mechanisms for transforming input states to output states.

We first consider what it means for a context to express a relation. Many statements can be expressed in terms of a subject, relation, and object (*s,r,o*). For instance, the statement *Miles Davis plays the trumpet* expresses a relation $F_r$, connecting the subject $s$ (*Miles Davis*) to the object $o$ (*trumpet*):

$$F_r(s) = o$$

We can then relate new subjects to objects: $F_r(\text{\textit{Jimi Hendrix}}) = \text{\textit{guitar}}$ and $F_r(\text{\textit{Elton John}}) = \text{\textit{piano}}$. $F_r$ is an inductive mechanism, from which statements about subject and object pairs can be obtained. We are interested in how a language model implements this abstraction.

**Affine LRE.** As a starting point, we look at the affine linear relational embedding (LRE) method developed by Hernandez et al. (2023). The authors are able to approximate the transformer's relational function $F_r(s)$ with the affine approximator $\text{LRE}(s)$, such that when applied to novel subjects, they reproduce LM object predictions.

The object retrieval function from a subject with a fixed relational context, $o = F_r(s)$, is modeled to be a first-order Taylor approximation of $F_r$ about a number of subjects $s_1 \ldots s_n$. For $i = 1 \ldots n$:

$$\begin{aligned}
F_r(s) &\approx F_r(s_i) + W_r(s - s_i) \\
&= F(s_i) + W_r s - W_r s_i \\
&= W_r s + b_r, \\
\text{where } b_r &= F_r(s_i) - W_r s_i
\end{aligned}$$

In order to obtain $W_r$ and $b_r$, the authors look towards the internals of the trained model. In a relational context, a model may rely heavily on a singular subject state to produce the object state. Accordingly, they borrow the Jacobian matrix of derivatives between vector representations of the subject and object to use as $W_r$. The bias $b_r$ is then the vector-valued offset between the transformed subject state $W_r\mathbf{s}$, and the true object state $\mathbf{o}$. For a fixed relation, they calculate the mean Jacobian and bias between $n$ enriched subject states $\mathbf{s}_1 \ldots \mathbf{s}_n$ and outputs $F_r(\mathbf{s}_1) \ldots F_r(\mathbf{s}_n)$:

$$W_r = \mathbb{E}_{\mathbf{s}_i}\left[ \left. \frac{\partial F_r}{\partial \mathbf{s}} \right|_{\mathbf{s}_i} \right]$$

$$b_r = \mathbb{E}_{\mathbf{s}_i}\left[ F_r(\mathbf{s}) - \left. \frac{\partial F_r}{\partial \mathbf{s}} \mathbf{s} \right|_{\mathbf{s}_i} \right]$$

This yields a relational approximator capable of transforming a $j^{\text{th}}$ layer subject state $x_s^j = \mathbf{s}$ [1] into the final object hidden state $x_o^L = \mathbf{o}$ [2]:

$$\mathbf{o} \approx \text{LRE}(\mathbf{s}) = \beta W_r \mathbf{s} + b_r$$

For instance, $\mathbf{s}$ may be the final $7^{\text{th}}$ layer subject token state, and $\mathbf{o}$ the $26^{\text{th}}$ layer object token state, e.g. the next-token prediction state.

**True Linear Encoding.** However, the affine LRE diverges from its namesake, the linear relational embedding introduced by Hinton (1986), in introducing a bias $b_r$ and scaling term $\beta$. While linearity

---

[1]Following Meng et al. (2022a), both this paper and the affine LRE focus primarily on middle-layer states.

[2]Note the introduction of a $\beta$ scaling parameter. The authors claim the affine LRE is limited by layer normalization: the $\mathbf{s}$ representation is normalized before contributing to $\mathbf{o}$, and $\mathbf{o}$ is normalized before token prediction by the LM head, resulting in a mismatch in the scale of the output approximation. We find that this conclusion is supported by empirical evidence from linear projections.

is assumed by Hernandez by calculating $W_r$ and $b_r$ from $\mathbb{E}_{s_i}$ over $i = 1 \ldots n$, choosing to use a Taylor series makes a weaker assumption, which is simply that $F_r$ is differentiable near the input state $s_i$. Under the assumption that the relation is not only differentiable, but linear, we would expect the following approximation to be valid:

$$\mathbf{o} \approx F'_r(s_i)\mathbf{s}$$

Then, under the hypothesis put forth by Hernandez et al. (2023), the linear approximation over $\mathbf{s}_1 \ldots \mathbf{s}_n$ within the same relation would be be the mean Jacobian, as seen in 4.4. If this approximation generalizes to novel subject-object pairs, it would indicate the presence of a linear map between the subject and object state vector spaces.

In the original LRE, concepts are represented as a learned vector in Euclidean space, while each relationship between concepts are learned matrices. Thus, the operation relating $(a^c, R^c)$ to a vector $b^c$ is the matrix-vector multiplication $R^c \cdot a^c$. In summary, the linear relational embedding developed by Paccanaro and Hinton (2001) has a purely multiplicative analogue in the transformer setting. This relation has previously been extended to affine approximations (Yang et al. 2021). However we find that the Jacobian approximator performs comparably, and in fact surpasses affine LRE on certain morphological relations.

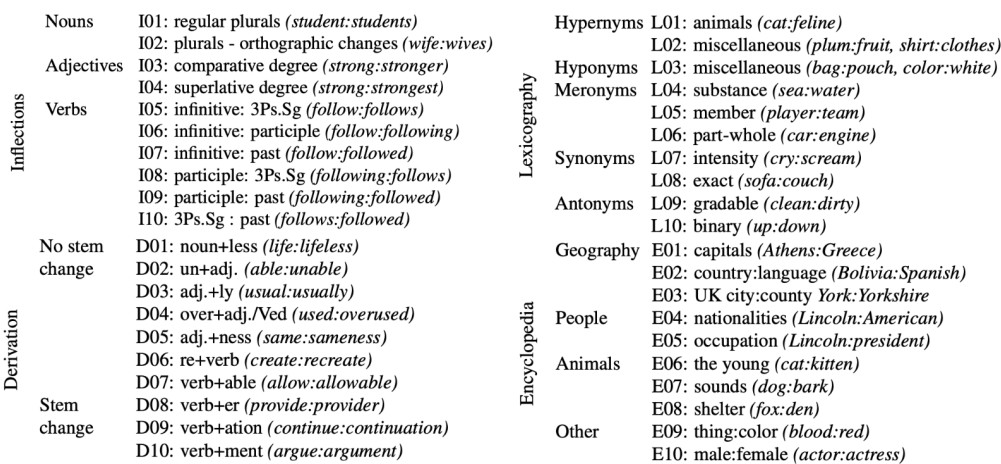

Figure 2: The BATS dataset structure from Gladkova et al. 2016

## 4.2 INTRODUCING NEW RELATIONS

From the cognitive perspective, analogy has traditionally been regarded as an inductive mechanism which makes comparisons between mental representations (Sternberg and Rifkin 1979, Gentner 1983). This makes analogy a special case of role-based relational reasoning (Holyoak 2012), and motivates the adaptation of analogical pairs to a relational setting. We choose to adapt the Bigger Analogy Test Set, also known as BATS. The Bigger Analogy Test Set was originally introduced to explore linguistic regularities in word embeddings by Gladkova et al. (2016). The dataset comprises forty different categories, spanning inflectional morphology, derivational morphology, encyclopedic knowledge, and lexical semantics. Each category is made up of fifty pairs of words sharing a common relation. The pairs are compiled from diverse manual and automated datasets, including WordNet, SemEval2012-Task2, Wikipedia, the Google Analogy Test Set, and a multimodal color dataset Fellbaum (1998); Jurgens et al. (2012); Mikolov et al. (2013); Bruni et al. (2012).

## 4.3 UTILIZING ICL

We adapt the relational pairs in BATS by introducing prompts which are compatible with each instance of the analogy. For instance, the [**verb+ment**] dataset comprises pairs of words with base ("fulfill") and derived ("fulfillment") forms. The prompt template given to the LM then elicits the

derived form from the base by a semantic relation: "To fulfill results in a ___________". This template is used across all pairs for a particular category.

Following the procedure in Hernandez 2023, we use 8 in-context learning (ICL) examples for 8 different subject-object prompts for each relation. This allows us to obtain a Jacobian from the model computation which is most likely to exhibit the desired linear encoding. For instance, we might extract the Jacobian for **[animal - youth]** with the following prompt:

```
The offspring of a dog is referred to as a puppy
The offspring of a sheep is referred to as a lamb
...
The offspring of a bear is referred to as a
```

We would like our approximations to generalize to unseen subject-object relations. Consequently, we omit the subject-object pairs used to construct the approximators from the testing pool. Additionally, we restrict evaluation of approximators to the pairs for which the LM computation is successful in reproducing the object in question: for both of the models we tested, GPT-J and Llama-7b, this is nearly all of the examples provided in BATS. See Appendix B for statistics on successful completion.

### 4.4 Evaluating the Jacobian

We are interested in how well each operator – the LRE, Jacobian, and Bias – are able to approximate the internal processes of the transformer. The approximated object tokens, after passing through the activation function in the decoder, should faithfully replicate the true LM output.

The original LRE is an affine approximation over a fixed relation. It has the subject hidden state $\mathbf{s}$ as input and the final object hidden state $\tilde{\mathbf{o}}$ as output:

$$\tilde{\mathbf{o}} = \text{LRE}(\mathbf{s}) = \beta W_r \mathbf{s} + b_r$$

Our variants isolate the components of the LRE in order to inspect their contribution to the approximation. In particular, if either the Jacobian or Bias approximator are able to successfully decode subject states comparably to the LRE, the affine representation put forth by Hernandez et al. (2023) may be unnecessary to approximate the model representation structure.

First, we define the Jacobian approximator, a multiplicative operation. This is the subject hidden state $\mathbf{s}$ multiplied by the mean Jacobian of *other subject-object pairs* to derive a final object state:

$$\tilde{\mathbf{o}} = \text{Jacobian}(\mathbf{s}) = W_r \mathbf{s}$$

Second, we define the Bias approximator, an additive operation. This approximator is adding $b_r$, the mean offset between $W_r \mathbf{s}$ and $\mathbf{o}$ for *other subject-object pairs*, to $\mathbf{s}$:

$$\tilde{\mathbf{o}} = \text{Bias}(\mathbf{s}) = \mathbf{s} + b_r$$

Following Hernandez et al. (2023), we define approximator faithfulness over a relation by the top-one token match rate for the approximation and the LM. When applied to unseen subjects $s$, the approximator output should match that of the LM. Denote the enriched subject state as $\mathbf{s}$, the transformer computation as $\mathbf{o}$, and the approximation as $\tilde{\mathbf{o}}$.

Then for token $t$ and decoder head $D$, we say an approximator is faithful if the top token approximation matches that of the LM:

$$\underset{t}{\arg\max} \, D(\mathbf{o})_t \overset{?}{=} \underset{t}{\arg\max} \, D(\tilde{\mathbf{o}})_t$$

## 5 Results

### 5.1 The Jacobian Faithfully Approximates Morphological Relations

We first evaluate relational approximators for the GPT-J model (Wang and Komatsuzaki 2021). We build approximators for likely subject hidden states (layers 3-9) and the final object state (layer 27)

through the process outlined above. We then evaluate the approximators four times over all forty relations, with randomized test prompts each iteration, and average the best performing approximation from each.[3] For the LRE, we use $\beta = 7$, which was found to be optimal for BATS. The Jacobian

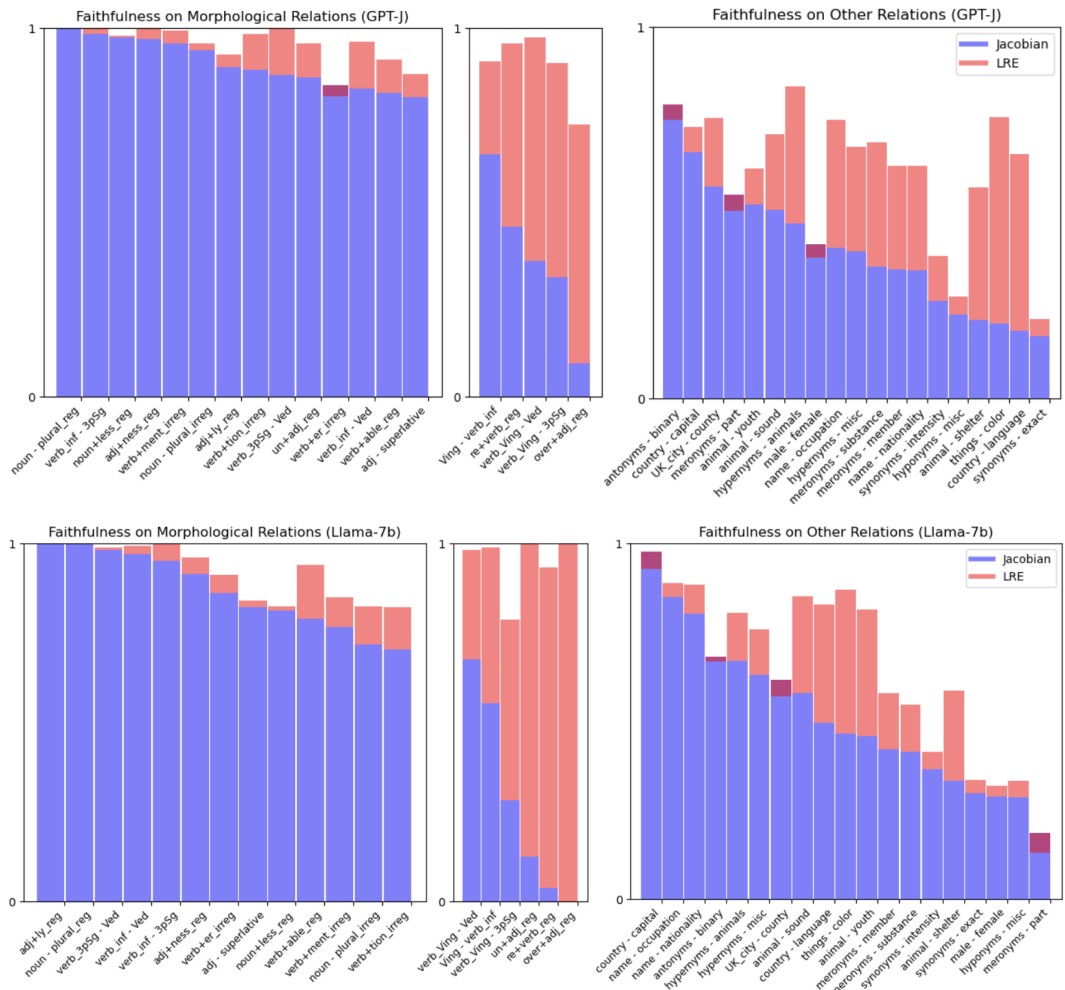

Figure 3: Comparing LRE and Jacobian faithfulness for morphological and other relations reveal many morphological relations are linearly approximable. With the exception of prefix and active form derivations, semantic and encyclopedic relations benefit far more from the affine LRE than morphological relations. Out of a range of subject layers (GPT-J: 3-9, Llama-7b: 4-16), the best performing approximation is averaged ($n = 4$).

approximator is able to achieve an average faithfulness of 90% across all 14 morphology relations which do not involve prefixes or an active base form, while the affine LRE achieves an average faithfulness of 95%. In contrast, the Jacobian approximator achieves an average faithfulness of 40% over non-morphological relations, while the affine LRE achieves an average faithfulness of 61%. This confirms the efficacy of the affine LRE found in Hernandez et al. (2023), while suggesting that some relations, e.g. morphological ones, may be encoded as truly linear.

The high faithfulness of the Jacobian shows that it is sufficient to approximate most morphological relations, but not that it is necessary. To show that the Jacobian is also necessary, we also compare against the Bias approximator, and find that Bias is unable to reproduce morphology faithfully.

---

[3]There were two relations which were not tested on, **[adj+comparative]** and **[antonyms-gradable]**. This was due to preprocessing difficulties.

While the Bias approximator is additive, the model might instead directly implement a linear combination to represent the final object state. As a consequence, we also compare against the TRANSLATION approximator, where the bias is formulated as $b = \mathbb{E}(o - \mathbf{s})$.[4] This approximator, from Hernandez et. al. 2023, calculates the direct offset of the subject and object hidden states, and is inspired by Merullo et al. 2023 and vector arithmetic. We find that without the Jacobian, bias approximations fail to approximate nearly all morphological relations, while successfully capturing some semantic and encyclopedic relations: the bias approximator achieves 67% faithfulness on **[things - color]**, while the TRANSLATION estimator attains 50% and 52% faithfulness on **[animal - shelter]** and **[hypernyms - misc]** respectively. This suggests that the multiplicative and additive mechanisms play complementary roles.

## 5.2 LLAMA-7B RESULTS

GPT-J utilizes parallel MLP and attention layers, unlike many other language models. Consequently, while these results show that the linear, multiplicative Jacobian is a faithful approximator for morphology in GPT-J, it is possible this observed linearity does not generalize. In order to ensure that our results hold across different models, we repeat the procedure for Llama-7b, which utilizes sequential attention and feedforward layers like most LLMs (Touvron et al. 2023). Llama-7b has 31 transformer layers: we sweep over subject layers 4-16. As seen in Figure 3 and 6, we obtain very similar results to GPT-J. Of particular note are the prefix and active form derivations: with the exception of **[un+adj_reg]**, the same morphological relations perform poorly under Jacobian approximation. This suggests that a similar encoding mechanisms exists across models.

## 5.3 CROSS-LINGUISTIC EVIDENCE

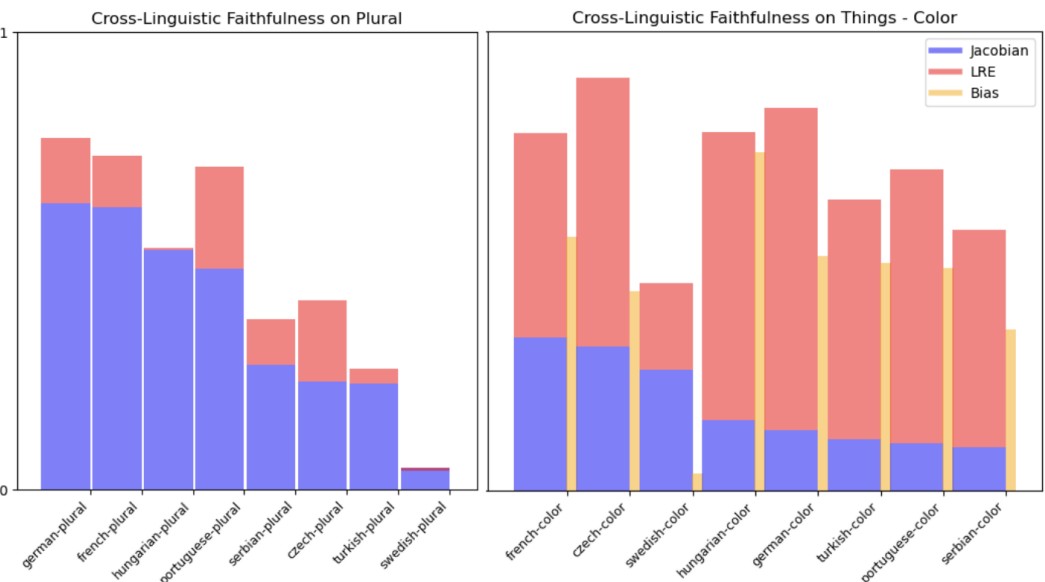

Figure 4: Evaluating languages present in Llama-7b reveal cross-typological linear encoding of morphology. It also supports the complementary role played by additive and multiplicative mechanisms.

We have shown that morphological relations in English are largely linearly decodable. However, the results may be limited to fusional-analytic languages with fewer unique affixes. Representations in other typological categories, such as aggluginative languages with rich morphology, may be encoded differently. For Llama-7b, we test Czech, French, German, Hungarian, Portuguese, Serbian, Swedish, and Turkish, the languages comprising portions of the training dataset. Hungarian and Turkish are both highly agglutinative. We create templates for two prototypical relations, one which involves morphology (**[plural]**) and one which does not involve morphology (**[things - color]**). We

---

[4]The results for TRANSLATION and Bias are available in the Appendix.

use the same methodology as above, sweeping over intermediate subject layer states and averaging the best performing approximations.

As seen in Figure 4, for **[plural]** the majority of the affine technique is approximable by the Jacobian, while **[things - color]** relies on an additive operation. Evaluating the average faithfulness as above, the affine LRE scores 68% on **[plural]** across four of these languages (German, French, Hungarian, Portuguese) while the Jacobian scores 56%. In contrast, the affine LRE scores 70% for **[things - color]** across all languages, whereas the Jacobian scores only 19%. The Bias approximator scores 45%, suggesting the affine approximation is primarily additive.

The evidence is indicative of a multiplicative linear relational embedding for morphological relations, independent of linguistic typology. Moreover, the high performance of Bias on color identification provides further evidence for complementary additive and multiplicative mechanisms for relational representation.

## 5.4 LINEAR PROJECTION

We produce interpretable object representations through linear projection in $\mathbb{R}^2$. Specifically, we use a basis of the bias vector and a random normalized vector, which has been orthogonalized with Gram-Schmidt to $b$. We project approximations s, $\beta W$s, $\beta W$s $+ b$ , as well as a calculated

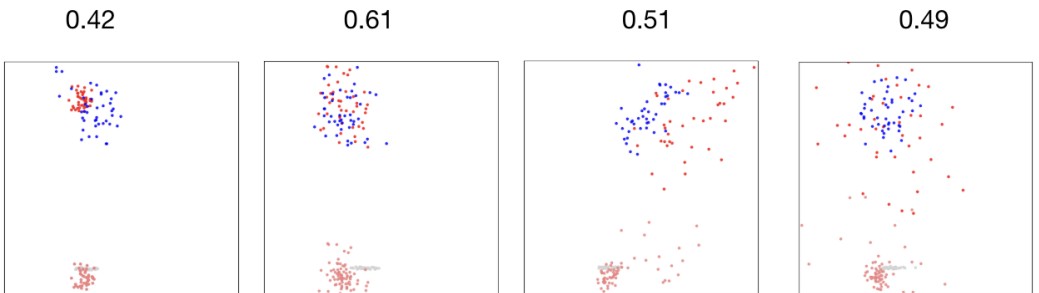

Figure 5: The $\{\perp, b\}$ subspace distances between $\beta W$s $+ b$ and **o** corresponds with the faithfulness scores displayed above. With $\beta$ values of 1, 3, 5, and 7, adjusting the hyperparameter is crucial for faithful approximation in the affine LRE.

hidden state for the correct object output **o**. These projections suggest $W$ is primarily responsible for transforming the underlying distribution to be geometrically similar to the output, while $b$ contributes the majority of movement in vector space. Through linear projection, we can validate that $\beta$ is necessary for recovering scale lost in layer normalization. In Figure 5, we see evidence that $\beta$ provides variance that was lost in layer normalization, as conjectured by Hernandez et al. (2023).

The term $b_r$ could be compared to the vectors used by Mikolov and many others, and the concept vector subsequently formalized by Park. However, the bias vector and the concept vector are not truly analogous. The bias term describes an offset from the transformed subject to the object: $b_r = \mathbb{E}(o - W_r \mathbf{s})$, not $b_r = \mathbb{E}(o - \mathbf{s})$. In practice, we find that bias and concept vectors are close in cosine similarity, and likely serve similar roles.

## 5.5 NON-STEMMED FORMS AND FAILURE CASES

The faithfulness metric is potentially a problematic choice for measuring morphology. High faithfulness scores on many morphological tasks can be achieved by reproducing a substring of the subject token. In this case, it is possible the Jacobian approximation is simply repeating a stemmed form of the subject token. Consequently, it would be agnostic to the derived form. This theory would make the high faithfulness of morphology more questionable.

However, the Jacobian produces many full morphological forms, which challenges this perspective. For instance, #25303 ' sadness' and #24659 ' continuation' faithfully replicate derived forms and are consistently reproduced by the Jacobian. For further evidence against this view, including correct, stemmed, and incorrect counts, see Table 1 and Table 3 in the Appendix.

There are two inflectional relationships the Jacobian failed to approximate as well over the tests performed, **[Ving - 3psg]** and **[Ving - Ved]**. One possibility is that transformations from the verb active form make the LM computation non-linear. For the majority of the relations on which the Jacobian achieves high faithfulness, the subject is the unmarked form, such as the verb infinitive or third person singular. There are also derivational prefix tasks for which the LRE, but not the Jacobian, faithfully approximates, **[re+verb]** and **[over+adj]**. A partial explanation for this phenomenon is that the object tokens "over" and "re" are idiosyncratically related to the subjects, unlike other relations. As seen in Table 2, this causes the vocabulary to contain fewer correct object hidden states, so transformations of the subject hidden state may not be an effective approximation.

### 5.6 IMPLICATIONS FOR CONCEPT THEORY

Morphological relations involve well characterized concepts between words. The Linear Relational Hypothesis formalized by Park et al. (2024) posits that directions in the representation space of a language model encode high-level concepts. However, contrary to expectation, we have found that morphological derivations are well-approximated with a multiplicative operation, and not by an additive operation. As can be seen in Figure 6 and Figure 7, both Bias and TRANSLATION results in faulty approximations for morphology. However, these results could still be compatible with the LRH. If morphology is encoded as a linear transformation, relations distinct from morphological paradigms (e.g. semantics specified at the lexical level) might continue to be represented by vectors.

We do not claim that morphological derivation is the only linguistic phenomenon which can be linearly approximated, or that all morphology is linearly approximable. Instead, we demonstrate the hidden states of base representations can be implicitly transformed to morphological derivatives, highlighting a surprising linearity present in many morphological relations.

## 6 CONCLUSION

In this work, we have adapted the Bigger Analogy Test Set to create a large novel testing dataset for relations, covering forty relations over morphological, factual, and semantic relations. We formulate the transformer equivalent of the linear relational embedding found in Paccanaro and Hinton (2001) more precisely to be equivalent to the Jacobian, and, surprisingly, find this approximator is able to model certain relations as well as the affine LRE. Returning to the affine method, we hypothesize that the Jacobian serves the role of extending a subject entity to alternative forms, and the bias term serves the role of shifting underlying concepts.

Through the approximation of language models, we arrive at a better understanding of their internal structure, which is crucial for controlling its outputs effectively. This ultimately has implications for many downstream applications of transformer language models, including as knowledge bases, dialogue agents, and as robust tools for inference and reasoning.

## 7 REPRODUCIBILITY STATEMENT

The approximation code is based on the LRE repository (Hernandez et al. 2023), and loads GPT-J and Llama-7b in half-precision. The code and dataset are available at {link}. Experiments were run remotely on a workstation with 24GB NVIDIA RTX 3090 GPUs using HuggingFace Transformers.

### ACKNOWLEDGMENTS

The research done here was supported by the National Science Foundation under award number #XXX. Any opinion, finding, or conclusion in this study is that of the authors and does not necessarily reflect the views of the National Science Foundation.

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

# A APPENDIX

## A.1 LIMITATIONS: SCALING

Our experiments were conducted exclusively on GPT-J and Llama-7b due to hardware constraints, which limited the scope of our evaluations. While GPT-J is a powerful language model, it has fewer parameters and lower computational complexity compared to state-of-the-art models. As a result, the linear decoding investigated here may not generalize to larger-scale models.

However, as transformer models share the same underlying architecture, smaller models serve as a likely proxy for studying the interpretability of transformer-based language models. Future work could build on these findings by scaling experiments to larger models, but we believe that architectural similarities allow our study to provide meaningful contributions to interpretability across transformers.

## A.2 EXPERIMENTAL LIMITATIONS

A key assumption being made is that the linear transformations observed here are employed in regular token prediction the same way that they are done in an explicit relational content. Based on existing literature in activation patching and editing (Geva et al. 2021), we believe that the hypothesis of subject enrichment being independent from contexts is supported. To determine this more thoroughly, further research could employ references to both base and derived forms in naturalistic contexts.

Additionally, unlike previous investigations of linear approximation, we did not investigate whether the faithfulness of the Jacobian approximation is associated with causality. Based on the prior work which successfully finds a relationship between these variables Hernandez et al. (2023), it is reasonable to believe these two measures are correlated, and that the internal structure of the model is revealed to be linear.

# B    SUCCESSFUL COMPLETIONS FOR GPT-J AND LLAMA-7B

Each relational prompt from the BATS dataset was evaluated for successful completion over 4 trials and averaged. As seen below, both models successfully complete the vast majority of objects.

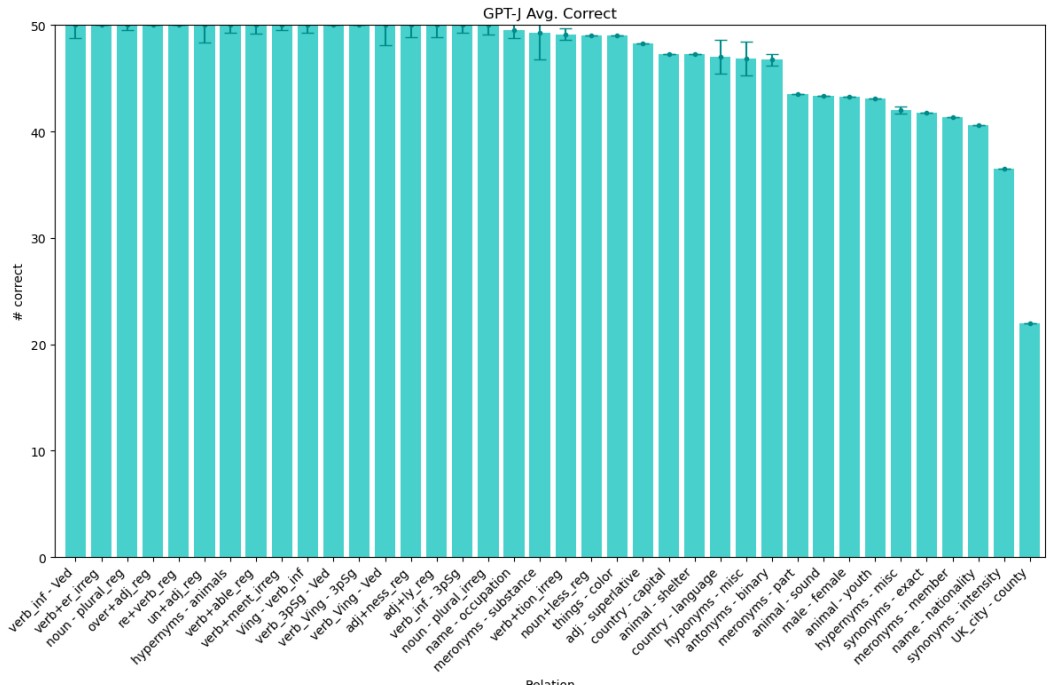

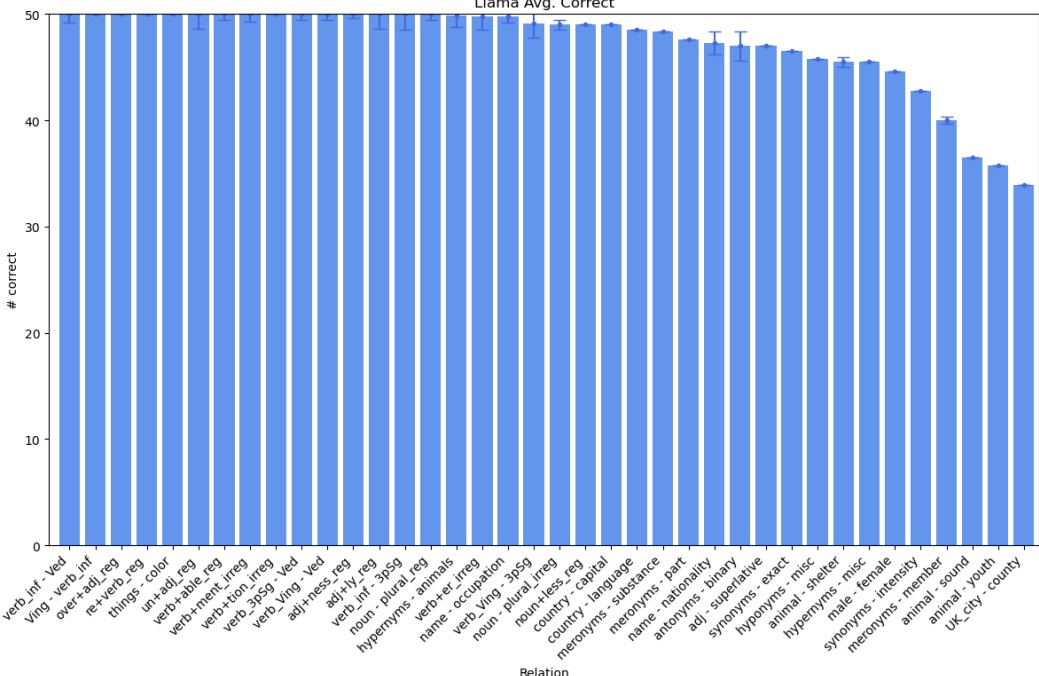

# C   EVIDENCE OF NON-STEMMED FORMS

All examples below are from GPT-J.

| Subject | Jacobian Top-3 |
|---|---|
| society | societies, Soc, soc |
| child | children, children, Children |
| success | successes, success, Success |
| series | series, Series, Series |
| woman | women, women, Women |
| righteous | righteousness, righteous, … |
| conscious | consciousness, conscious, … |
| serious | seriousness, serious, serious |
| happy | happiness, happy, happy |
| mad | madness, mad, being |
| invest | investment, invest, investing |
| amuse | amusement, amuse, amusing |
| accomplish | accomplishment, accomplish, … |
| displace | displacement, displ, dis |
| reimburse | reimbursement, reimburse, reimb |
| globalize | globalization, global, international |
| install | installation, install, Installation |
| continue | continuation, continu, contin |
| authorize | authorization, Authorization, … |
| restore | restoration, restitution, re |
| manage | manager, managers, manager |
| teach | teacher, teachers, teach |
| compose | compos, composer, composing |
| borrow | borrower, lender, debtor |
| announce | announcer, announ, ann |

Table 1: **[noun_plural], [verb+er], [verb+ment], [adj+ness], [verb+tion]** Selected examples of full subject tokens demonstrate that relational Jacobian approximation is able to capture irregular morphology effectively, and does not merely reproduce stemmed subjects.

| Relation | # Unique |
|---|---|
| **un+adj** | **7** |
| **over+adj** | **4** |
| **re+verb** | **15** |
| name - nationality | 13 |
| animal - shelter | 18 |
| synonyms - intensity | 35 |
| verb+able | 47 |
| noun - plural | 47 |

Table 2: The number of unique start tokens for correct objects across selected BATS relations. Less unique start tokens correspond to less injective mappings from subject to object, which may be harder to approximate linearly.

| Correct | Stemmed | Incorrect |
|---|---|---|
| 42 | 0 | 0 |
| 23 | 11 | 9 |
| 7 | 35 | 6 |

Table 3: Correct, stemmed, and incorrect suffix counts for **[noun_plural]**, **[verb+tion]** and **[adj+ness]** from the top prediction of a fixed layer Jacobian approximation further suggests consistent linear encoding beyond stemmed forms.

# D    BIAS RESULTS DEMONSTRATE $W$ NECESSITY

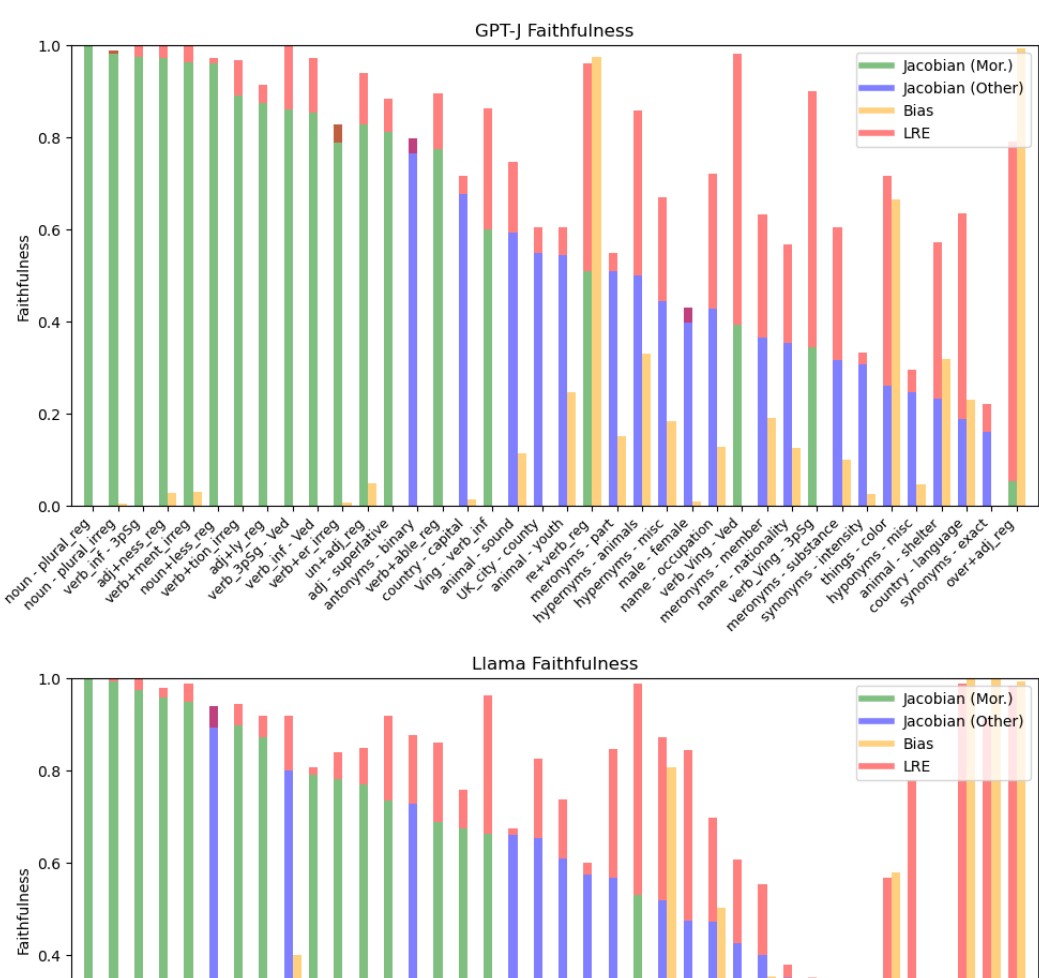

Figure 6: A comparison of linear and affine approximators against the bias approximator demonstrate the necessity of $W$ in the LRE. The bias approximator successfully models some relations, but only when the gap between the Jacobian and LRE is large, mostly in semantic and encyclopedic relations. This suggests the operations play complementary roles.

# E TRANSLATION RESULTS DEMONSTRATE $W$ NECESSITY

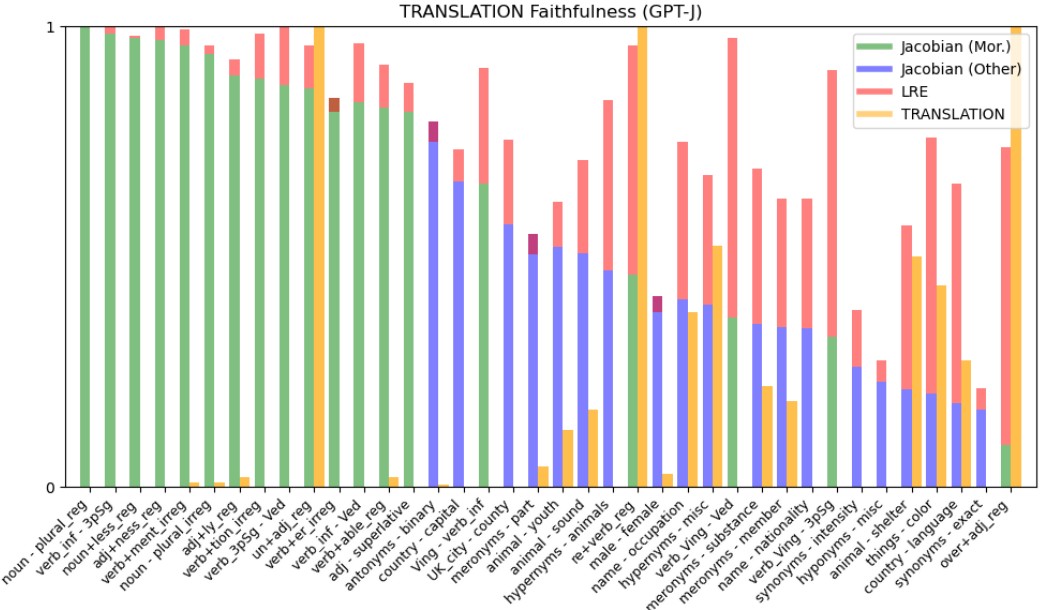

Figure 7: A comparison of linear and affine approximators against the TRANSLATION approximator, $b = \mathbb{E}(o - \mathbf{s})$. Like Bias, the TRANSLATION approximator is generally successful when the gap between the Jacobian and LRE is large, mostly in semantic and encyclopedic relations.

