# OpenReview forum: "Linear Relational Decoding of Morphological Relations in Language Models"
_ICLR.cc/2025/Conference — Submitted to ICLR 2025_

### Official Review · Reviewer_9Gx3 · 2024-11-04

**Soundness:** 3
**Presentation:** 2
**Contribution:** 3
**Rating:** 3
**Confidence:** 3

**Summary:**

The paper uses that morphological relations exist in a linear subspace after looking at a Taylor-expansion of the neural network. (I am not entirely sure which Taylor expansion; see below). The claim to novelty of the paper is that linear relationships have been tested for other types of linguistic relations, but for morphological. So, as an example, if I have the complex word "uncomputable" I want to break it down into "un"-"compute"-"able" and show that each piece, when encoded as a real vector, is something I can sum to get the entire word. The experimental results focus on translation of morphologically complex words.

**Strengths:**

The strength of the paper stems from its focus on a less well-reached topic. I appreciated an interpretability paper that focuses on morphology. I think if the method were explained better, it would be a very interesting paper.

**Weaknesses:**

The primary weakness of the paper is its poor technical exposition. I've phrased most of this as questions below. Another weakness is the poor rendering of the plots. It's very hard to read the plots in Figure 5. Also, many of the equations are not labeled making them hard to reference in the review. The background section, 2.1. is also hard to read.

**Questions:**

Most of the questions are about section 2.
* What is F? To talk about Jacobians, I imagine it has to be a vector-to-vector map.
* At what point is the Jacobian computed? W is defined as a function, but treated as a matrix?
* What do the authors deal with tokenization?
* What are the examples talked about in section 2.4?
* Is unbolded o the same as bolded o? It sort of looks like it

---

> ### Author Response · Authors · 2024-11-22
>
> Thank you for the detailed questions on the methodology and problem statement! We appreciate your note on the potential for the work. We have revised the notation throughout, aiming to focus on our key contributions and introduce technical details more seamlessly. We recognize that the paper introducing the affine LRE (Hernandez) is unlikely to be read even by domain experts, and so we have moved relevant background to section 2. We also answer your questions below.
>
> > What is F? To talk about Jacobians, I imagine it has to be a vector-to-vector map.
>
> Following Hernandez 2023, we define $F = F_r$ as the transformer prediction of the object under an invariant subject-object relation. Because the relation is fixed while approximating $F_r$, we assume that the object prediction depends on a single subject token. For instance, consider the statement, "the plural form of datum is data". In this case, $F_r(\text{datum}) = \text{data}$.  When we approximate $F_r$, we use the Jacobian of a vector-to-vector map, from the last token of an intermediate subject state to the final layer of the token prediction. This reliance on a singular state is a plausible outcome from the attention mechanism. By testing on a large novel dataset, we demonstrate empirically that this reliance is especially strong for morphological relations, with linear approximation outperforming the affine method introduced by Hernandez over certain morphological relations.
>
> Our use of $F$ is only for notational convenience, which unfortunately introduces confusion in broader usage. We have revised the paper to exclusively refer to $F_r$.
>
> > At what point is the Jacobian computed? W is defined as a function, but treated as a matrix?
>
> As mentioned in section 2.2, the Jacobian is taken from the transformer computation over fixed relational contexts in order to build a relational approximator. W is defined in section 2.4 as the average of Jacobians from contexts with different subjects, and is able to generalize to other prompts under the same relation. For instance, the average of the Jacobians from the transformer computations on "the offspring of a dog is a (puppy)" and "the offspring of a sheep is a (lamb)" is used to approximate the transformer computation for "the offspring of a bear is a (cub)".
>
> > What do the authors deal with tokenization?
>
> As outlined in 2.1, Meng (2022) and Geva (2023) find that the last subject token in middle layers are strongly casual, so that our approximator looks solely at the middle layer hidden states of those tokens. This leads to high faithfulness over our outputs. As outlined in 4.5, due to the tokenization of outputs we only consider single-token outputs, and address threats to validity arising from our use of the faithfulness metric in the Appendix.
>
> > What are the examples talked about in section 2.4?
>
> In 2.4, the examples refer to the subjects in the relational prompts from which the Jacobian is derived. The following equation block describes the affine formulation of the LRE used in Hernandez; further down, we introduce the linear form which we use.
>
> > Is unbolded o the same as bolded o? It sort of looks like it
>
> As defined at the bottom of page 3 in 2.4, $\textbf{o}$ is the object hidden state at the final layer L. Unbolded o refers to a type-level representation of the object, which is utilized only in the starting assumption for the relational function $o = F_r(s)$.

---

> ### Author Response · Authors · 2024-11-30
>
> We hope this message finds you well. As we approach the conclusion of the discussion period, we would appreciate a reexamination of the paper. We have made updates to the notation and refined the technical exposition throughout the paper to improve clarity and precision. Any further feedback would be appreciated.

---

> > ### Comment · Reviewer_9Gx3 · 2024-11-30
> > **Thanks**
> >
> > I'll take a look! Thanks for pinging me.

---

### Official Review · Reviewer_c3s8 · 2024-11-05

**Soundness:** 2
**Presentation:** 3
**Contribution:** 2
**Rating:** 3
**Confidence:** 4

**Summary:**

The paper explores how transformer-based language models approximate morphological transformations using linear transformations. The study finds that for morphological relations, a simple Jacobian (linear) transformation effectively captures the relationship between base and derived word forms (e.g., noun to plural, verb to noun). This linear approximation achieves high accuracy on morphological tasks from the Bigger Analogy Test Set and shows consistency across different language models, suggesting that morphology is encoded in transformers as a linear transformation.

**Strengths:**

- Demonstrates that morphological relations in language models are highly linearly approximable, providing insights into model structure and interpretability.
- Validates findings across different language models, indicating robustness of the method.
- Uses the Bigger Analogy Test Set, covering a wide range of morphological and linguistic relations for validation across a range of morphological paradigms.

**Weaknesses:**

- The method's reliance on linear transformations is highly likely to be limited to English, a fusional-analytic language. Most language are highly non-linear in morphological transformations and thus paper has to state its linguistic limitations and scope before claiming a general theory of any kind.

**Questions:**

-

---

> ### Author Response · Authors · 2024-11-22
>
> Thank you for the assessment of the strengths and weaknesses of our paper. We are pleased to find that you see our paper as yielding insight into model structure and interpretability. In response to your primary criticism, we have run additional experiments evaluating linear approximation in a multilingual setting. For Llama-7b, we test Czech, French, German, Hungarian, Portuguese, Serbian, Swedish, and Turkish, the languages identified as part of the training dataset. Hungarian and Turkish are both highly agglutinative. We create templates for two prototypical relations, one involving morphology (plural) and one which does not involve morphology (things-color). We use the same methodology as with BATS, sweeping over intermediate subject layer states and averaging the best performing approximations.
>
> We show that for the average faithfulness on plural forms, the LRE scores 68% across four of these languages (German, French, Hungarian, Portuguese) while the Jacobian scores 56%. In contrast, the LRE scores 70% for color identification across all languages, whereas the Jacobian scores only 19%. The Bias approximator scores 3-4% for reproducing plurals, while scoring 45% for color identification.
>
> This experiment is strongly indicative of linearity in morphological transformations, independent of linguistic typology. Moreover, the performance of the Bias on color identification provides evidence for the existence of complementary *additive* and *multiplicative* mechanisms mapping from subject to object states, as referenced within the paper.

---

> ### Author Response · Authors · 2024-11-30
>
> Hello Reviewer c3s8,
> We have a short time left in the discussion period, so please let us know if we have addressed your concern with our multilingual experiment. This was conducted specifically to address your feedback, so any additional updates or comments would be appreciated.
>
> Thank you!

---

### Official Review · Reviewer_C6Uh · 2024-11-06

**Soundness:** 3
**Presentation:** 2
**Contribution:** 2
**Rating:** 5
**Confidence:** 2

**Summary:**

This work (building off Hernandez et al) shows that linear approximations are sufficient to encode morphological relations from an object to subject state. The authors find strong evidence that the Jacobian works for morphological relations, but performs worse than the LRE for other kinds of relations.

They further characterize the roles of Jacobian and bias (in relation to LRE), and help define structures behind output generation in LLMs better.  The paper performs these experiments on a novel adaptation of the Bigger Analogy Test Set, wherein forty distinct relations are represented.

**Strengths:**

* The experiments are very thorough and well done. The authors spend a lot of effort trying to understand latent justifications to the results they obtain. The mathematical justifications of the research are well motivated, with a good set of references to back the claims.
* This is an important area of research in latent understanding of LLMs.

**Weaknesses:**

* There seem to be multiple contributions of the paper, some being less important than others. This makes it hard to clearly understand 1-2 important takeaways from the work. As examples, these takeaways can be Jacobian approximations, and role of bias and Jacobian in LRE. The rest should be condensed into a single para and maybe kept as smaller side-contributions.
* The methodology of adaptation of Bigger Analogy Test Set should be clearly explained. I am assuming this is an important aspect of the work?
* Is there any reason the authors didn’t try experiments with larger models (greater than 10B params)? Learnings might be significantly different.
* The presentation and in general, representations of facts isn’t done in an ideal manner.
    * Fig 3 and 4 are not interpretable. Please enlarge the image, or in general, feel free to just present the most relevant facts.
    * A lot of the abbreviations like LRE and BATS are not defined. It’s clear what they represent if you read the paper properly, but a definition is warranted.

**Questions:**

See weaknesses,

---

> ### Author Response · Authors · 2024-11-22
>
> Thank you for providing specific suggestions for improvement and evaluating our paper as a whole. We are pleased to find that you find our experiments very thorough and well done. We have revised both the content and presentation of the paper to reflect your concerns.
>
> > There seem to be multiple contributions of the paper, some being less important than others. This makes it hard to clearly understand 1-2 important takeaways from the work.
> > 	The presentation and in general, representations of facts isn’t done in an ideal manner.
> > Fig 3 and 4 are not interpretable. Please enlarge the image, or in general, feel free to just present the most relevant facts.
> > A lot of the abbreviations like LRE and BATS are not defined. It’s clear what they represent if you read the paper properly, but a definition is warranted.
>
> We agree that the paper has multiple contributions which vary in importance. We have removed Figure 4 and other less relevant sections. We explicitly list the takeaways in 5.1 and 5.3. We have also changed the presentation of Figure 5, and revised the focus of the Introduction, Related Work, and Background sections to emphasize the efficacy of the Jacobian, and the roles played by the bias and Jacobian in the LRE.
>
> > The methodology of adaptation of Bigger Analogy Test Set should be clearly explained. I am assuming this is an important aspect of the work?
>
> We have revised Introducing New Relations (4.2) to motivate the use of analogical datasets for relational approximation. We also added an example prompt template in Utilizing ICL (4.3), and provide more specific instances of relations throughout the paper.
>
> > Is there any reason the authors didn’t try experiments with larger models (greater than 10B params)? Learnings might be significantly different.
>
> Our experiments were conducted exclusively on GPT-J (6 billion parameters) and Llama (6.7 billion parameters) due to hardware constraints. These are not small models, and are able to  reproduce true object predictions consistently, which suggests a learned mechanism. Many interpretability papers use models in the same size or smaller: for instance, GPT-2-XL is 1.5B. We believe that the models we used serve as an effective proxy for studying the structure of larger models.  However, we agree there may be differences which arise from scaling; as a result, we have extended the limitations section to address the model sizes used in the experiment.

---

> > ### Comment · Reviewer_C6Uh · 2024-11-24
> >
> > Thanks for the response. The changes described are a quite large, and would alter the paper significantly. Hence, my score remains the same.

---

### Author Response · Authors · 2024-11-22

We would like to thank all reviewers for providing detailed areas for improvement. We have updated our submission to address their concerns, and believe that their feedback has substantially strengthened the contributions of the paper. Here is a summary of the changes made:

**Added Experiment: Cross Linguistic Evidence (5.3, Figure 4)**

We test eight languages found in the training data for Llama-7b, and find results which support linear relational embeddings for morphological changes cross-typologically. We also find additional evidence that the additive (Bias) and multiplicative (Jacobian) mechanisms play complementary roles in the LRE.

**Added Limitations: Scale (Appendix)**

**Modified Figure 5 to render points instead of subjects**

**Revised all sections**
 - Related Work (previously Section 5) was moved to Section 2
 - Justify BATS adaptation (4.2)
 - Background (3) now provides a thorough explanation of the affine LRE
 - Introduction (1) now provides further justification for studying relational approximation
 - Added example prompt template to 4.3
 - Removed Underlying Mechanisms (5.3)
 - Removed In-Context Learning (2.3)
 - Removed linear projections (previous Figure 4)

We welcome further suggestions and comments, and hope that reviewers consider adjusting their score as they see fit.

---

### Meta-Review · Area_Chair_52sx · 2025-01-03

**Metareview:**

This paper applies the Linear Relational Embedding (LRE) approach, previously developed by Hernandez et al. (2024) for factual/encyclopedic relations, to morphological relationships in language models. The authors also separate the contributions of bias and Jacobian approximators to understand their respective roles. They show that linear approximations are sufficient to encode morphological relations, i.e. that morphological transformations can be approximated using just the linear component (Jacobian) of the LRE, without requiring the bias term.

It seems that the main issue the reviewers had with the paper was clarity, including figures, notation, and technical exposition. During the rebuttal, the authors did a fairly big rewrite of the submission and also validated their findings in multilingual experiments on languages with different morphological structure. My feeling is that the paper is greatly improved, but it now needs to be re-reviewed.

The review process would need to ensure that: a) the issues with technical exposition have been adequately addressed, b) the paper's relationship to Hernandez's work is clearly articulated and contributions differentiated beyond application to morphology, and c) the insights from separating bias and Jacobian approximators are empirically grounded with adequately supported conclusions.

**Additional Comments On Reviewer Discussion:**

The reviewers primarily focused on clarity issues. Reviewer C6Uh highlighted the need to clarify key contributions and improve figure presentation. Reviewer c3s8 raised concerns about cross-linguistic generalization. Reviewer 9Gx3 identified multiple technical exposition issues requiring clarification. The authors addressed these concerns through additional experiments and explanations, producing a substantial revision of their original submission.

---

### Decision · Program_Chairs · 2025-01-22

Reject